# Inflammation and Fibrosis in Patients with Non-Cirrhotic Hepatitis B Virus-Associated Hepatocellular Carcinoma: Impact on Prognosis after Hepatectomy and Mechanisms Involved

**Yan Li [1], Jing-Fei Zhao [1], Jie Zhang [1], Guo-Hua Zhan [1], Yuan-Kuan Li [1], Jun-Tao Huang [1], Xi Huang [2] and Bang-De Xiang [1,\*]**

[1]  Department of Hepatobiliary Surgery, Guangxi Medical University Cancer Hospital, Nanning 530021, China
[2]  The First Clinical School of Guangxi Medical University, Nanning 530021, China
\*  Correspondence: xiangbangde@gxmu.edu.cn

**Abstract: Background:** We investigated whether the degree of inflammation and fibrosis in para-carcinoma tissue can predict prognosis of patients with non-cirrhotic hepatitis B virus (HBV)-associated hepatocellular carcinoma (HCC) after hepatectomy. We also explored the mechanisms through which inflammation and fibrosis might affect prognosis. **Methods:** Clinicopathological data were retrospectively analyzed from 293 patients with non-cirrhotic HBV-associated HCC who were treated at our institution by curative resection from 2012 to 2017. Based on the Scheuer score system, patients were classified into those showing mild or moderate-to-severe inflammation and fibrosis. Rates of overall and recurrence-free survival were compared between the groups using Kaplan–Meier curves, and survival predictors were identified using Cox regression. Using tumor and para-tumor tissues from independent samples of patients with non-cirrhotic HBV-associated HCC who were treated at our institution by curative resection from 2018 to 2019, we performed next-generation sequencing and time-of-flight cytometry (CyTOF) to examine the influence of inflammation and fibrosis on gene expression and immune cell infiltration. **Results:** In the analysis of the 293 patients, those with mild inflammation and fibrosis showed significantly better overall and recurrence-free survival than those with moderate-to-severe inflammation and fibrosis. Multivariate Cox regression confirmed that moderate-to-severe inflammation and fibrosis were independent risk factors for worse survival. RNA sequencing and CyTOF showed that more severe inflammation and fibrosis were associated with stronger invasion and migration by hepatocytes. In cancerous tissues, the biological processes of cell proliferation were upregulated, the signaling pathways promoting tumor growth were activated, the proportion of Th17 cells promoting tumor progression was increased, and CD8+ T cells expressed higher levels of PD-L1. In para-cancerous tissues, biological processes of immune response and cell chemotaxis were downregulated, and the proportion of tumor-killing immune cells was decreased. **Conclusion:** Worse inflammation and fibrosis in non-cirrhotic HBV-associated HCC is associated with worse prognosis, which may reflect more aggressive tumor behavior and an immunosuppressed, pro-metastatic tumor microenvironment.

**Keywords:** inflammation; fibrosis; hepatocellular carcinoma; hepatitis B virus; RNA sequencing; CyTOF





## 1. Introduction

Hepatocellular carcinoma (HCC) is a devastating cancer that accounts for the sixth largest number of cancer deaths worldwide [1], and rising HCC incidence places a heavy burden on healthcare systems [2]. The pathogenesis of HCC is complex and unclear [3], but the disease, in most patients in China, can be linked to chronic liver disease, especially chronic infection with hepatitis B virus (HBV) [4]. After HBV infects hepatocytes, its covalently closed circular genome can persist there for the individual's lifetime, permanently

increasing risk of liver injury and progression to HCC [5–7]. In addition, the superposition of external factors can also lead to more severe liver injury. Previous studies have shown that ongoing alcohol consumption was associated with liver fibrosis progression, and that ethanol intake was also a risk factor for liver injury after HBV infection [8].

Chronic HBV-associated hepatitis involves not only inflammation but also fibrosis, and the two reinforce each other causing progression to liver cirrhosis and HCC in a significant proportion of patients [9–11]. In contrast, HCC precedes cirrhosis in 7–54% of patients [12–14], and the mechanism by which liver injury following HBV infection affects the prognosis of such non-cirrhosis HCC patients needs to be further investigated.

Most previous studies of such patients have considered how inflammation or fibrosis on their own influence prognosis [15,16]. We wondered, instead, whether the interaction between inflammation and fibrosis in para-cancerous tissues might affect prognosis. Therefore, we compared survival between non-cirrhotic HBV-associated HCC patients who had more or less severe inflammation and fibrosis based on the Scheuer system. We also used RNA sequencing and time-of-flight cytometry (CyTOF) to begin to elucidate how inflammation and fibrosis might influence prognosis.

## 2. Method

The study was approved by the Institutional Ethics Committee of Guangxi Medical University Cancer Hospital (approval LW2021096), and written informed consent was obtained from each participant. The study was registered in the Chinese Clinical Trial Registry as ChiCTR2100053183.

### 2.1. Influence of Inflammation and Fibrosis on Survival

#### 2.1.1. Patients

Patients with non-cirrhotic HBV-associated HCC who underwent hepatectomy from January 2012 to December 2017 at Guangxi Medical University Cancer Hospital (Nanning, China) were consecutively enrolled. Inclusion criteria were the following: (1) patients were diagnosed with HCC based on pathology according to World Health Organization criteria; (2) R0 hepatectomy was performed, defined as complete macroscopic removal of the tumor, negative resection margins, and no detectable intra- or extrahepatic metastatic lesions; (3) preoperative imaging indicated no distant metastasis; (4) patients had a preoperative Child–Pugh score of A or B; and (5) hematologic testing at one week before surgery was positive for HBV surface antigen.

The following exclusion criteria were applied: (1) emergency operation because of rupture and hemorrhage of liver cancer; (2) prior anticancer treatment, such as transarterial chemoembolization or radiation; (3) postoperative pathology findings suggesting that para-carcinoma liver tissue had become cirrhotic; (4) incomplete medical records; (5) perioperative death; (6) follow-up for fewer than three months; or (7) history of other malignancies.

#### 2.1.2. Surgery and Patient Assessment

All procedures were performed by experienced surgeons in the Department of Hepatobiliary Surgery at our institution using standard techniques. Liver resection specimens were retrospectively assessed by two experienced pathologists, who were blinded to patient demographic characteristics and clinical outcomes. The degree of inflammation and fibrosis were assessed using the Scheuer system [17] (Table 1), and patients were divided into those showing mild inflammation and fibrosis ($G \leq 2$ and $S \leq 2$) or moderate-to-severe inflammation and fibrosis ($G$ or $S > 2$) [18,19].

Preoperative alpha-fetoprotein (AFP), liver function levels, and blood test results within seven days before surgery were extracted from the hospital database. Other clinical information was extracted from standardized pathology reports, such as the number of tumors, tumor size, and surgical margins. Tumor size was defined as the maximum diameter of the largest tumor in resected specimens. Tumors were classified as grade I (well

differentiated), grade II (moderately differentiated), or grade III–IV (poorly differentiated) according to the Edmondson–Steiner criteria [20].

**Table 1.** Criteria of the Scheuer system.

| | Activity of Inflammation (G) | | Degree of Fibrosis (S) | |
|---|---|---|---|---|
| **Grade** | **Portal/Periportal Activity** | **Lobular Activity** | **Stage** | **Fibrosis** |
| 0 | None or minimal | None | 0 | None |
| 1 | Portal inflammation | Inflammation but no necrosis | 1 | Enlarged, fibrotic portal tracts |
| 2 | Mild, piecemeal necrosis | Focal necrosis, acidophilic bodies | 2 | Periportal or portal–portal septa, but intact architecture |
| 3 | Moderate, piecemeal necrosis | Severe focal cell damage | 3 | Fibrosis with structural distortion, but no obvious cirrhosis |
| 4 | Severe, piecemeal necrosis | Damage includes bridging necrosis | 4 | Probable or earlier cirrhosis |

Adapted from ref. [17].

### 2.1.3. Antiviral Treatment

All patients received perioperative antiviral therapy [21], and patients were advised to take antiviral therapy indefinitely unless side effects were unacceptable, as per routine practice at our institution. All patients showed HBV DNA below the limit of detection (100 IU/mL) within 2 months after surgery.

### 2.1.4. Follow-Up

Follow-up was performed by telephone or on an outpatient basis at one month after surgery, then every three months for the rest of the first year, and every six months thereafter. Follow-up ranged from 4 to 82 months, with no patients lost to follow-up. Follow-up visits included physical examination, liver function tests, quantitation of serum AFP and HBV DNA, abdominal ultrasonography, and computed tomography or magnetic resonance imaging.

### 2.1.5. Survival Assessment

Recurrence-free survival (RFS) and overall survival (OS) were the endpoints of the study. Survival time was calculated from the date of operation until the date of recurrence and/or death, or until the last follow-up visit in May 2021. Recurrence was defined as detection of a new lesion (1) in the liver during follow-up by ultrasonography, computed tomography, and/or magnetic resonance imaging, showing "fast in and fast out" findings typical of HCC [22]; or (2) in the lungs, bone, brain, abdominal cavity or other sites based on at least one of the imaging modalities described above.

### 2.1.6. Statistical Analysis of Survival Data

Statistical analyses were performed using SPSS for Windows 23.0 (IBM, Chicago, IL, USA). Intergroup differences in categorical data were assessed for significance using the Chi-squared or Fisher's exact test. Kaplan–Meier curves were used to estimate cumulative OS and RFS rates, and curves for the two groups were compared using the log-rank test. Differences associated with $p < 0.05$ were considered statistically significant. Uni- and multivariate Cox regression was used to identify independent predictors of OS and RFS, based on hazard ratios (HRs) and associated 95% confidence intervals (CIs).

Since patients were grouped based on their clinicopathological characteristics, we considered it highly likely that the two groups had baseline differences that might confound

our analysis. Therefore, we balanced these potential baseline differences by matching patients in the two groups 1:1, based on propensity scoring with a caliper width of 0.1 [23].

### 2.2. Influence of Inflammation and Fibrosis on the HCC Transcriptome

#### 2.2.1. Sample Collection

Carcinoma and para-carcinoma tissues that had been surgically resected from 52 patients with non-cirrhotic HBV-associated HCC were analyzed using RNA sequencing. Patients underwent resection at our institution between 2018 and 2019, and they satisfied the same inclusion and exclusion criteria as the patients in Section 2.1.1. The resected tissues had been frozen in liquid nitrogen and stored at −80 °C until analysis. Inflammation and fibrosis were classified as mild in half the subjects, or as moderate-to-severe in the other half.

#### 2.2.2. Preparation of cDNA Library and RNA Sequencing

Total RNA was extracted from tissue samples using HiPure Universal RNA Mini Kit (Guangzhou Magen Biotechnology, Guangzhou, China) according to the manufacturer's instructions, and treated with deoxyribonuclease to remove DNA. RNA integrity was accessed using an Agilent Bioanalyzer 2100 (Agilent, Palo Alto, CA, USA), and the optical density (OD) value of RNA was assessed using an Epoch2 spectrophotometer (Bio Tek, Vermont, NE, USA).

Cytoplasmic and mitochondrial ribosomal RNA was removed from total RNA extracts using the Ribo-Zero™ Magnetic Gold Kit (catalog no. MRZG12324, Illumina, San Diego, CA, USA). Fragmentation buffer was added to the treated RNA. The RNA was first cleaved randomly into template fragments 200–500 bp long by heating in the presence of divalent cations. Second, the initial cDNA strand was synthesized by reverse transcription using random hexamers, then mixed with ribonuclease H, DNA polymerase IdNTP (with dUTP instead of dTTP), and fragmentation buffer. The complementary strand cDNA was synthesized, then purified using AMPure XP beads (Beckmen Coulter Inc, Brea, CA, USA) to purify the double-stranded cDNA. The uracil-containing strand of cDNA was degraded using uracil-specific excision reagent. A poly(A) base was added to the 3′ end, and the fragment was ligated to universal Illumina sequencing adaptor. Products of self-ligation and incomplete ligation were removed, then AMPure XP beads were added to select fragments of different sizes. Bridge PCR was carried out using universal primers complementary to the ligation sequence, and the amplicons were purified and used to prepare the final double-stranded cDNA library.

#### 2.2.3. Next-Generation High-Throughput RNA Sequencing

The concentration of the sequencing library was measured using the Qubit 2.0 Fluorometer dsDNA HS Assay (Thermo Fisher Scientific, Waltham, MA, USA). The distribution of cDNA fragments in the library was determined using the Agilent BioAnalyzer 2100 by DNA agarose gel electrophoresis. Finally, the cDNA library was sequenced using the $2 \times 150$ paired end sequencing protocol (Illumina) on a Hiseq2000 high-throughput sequencer (Illumina). Raw sequencing data were obtained in FASTQ format.

#### 2.2.4. Bioinformatic Analysis

FastQC software (https://www.bioinformatics.babraham.ac.uk/projects/fastqc/, accessed on 1 May 2021) was used for quality control of the raw data of sequencing. The sequence alignment software Hisat [24] was used to compare the quality-controlled raw data with the human reference genome sequence "gencode v 19". The gene expression levels were quantified in terms of fragments per kilobase of transcript per million (FRKM). Log$_2$-transformed FRKM values were used for further analyses, except for extraction of differentially expressed genes (DEGs), which was performed using the package edgeR (version 3.36.0) [25] in the R software (version 4.1.2) (https://www.r-project.org/, accessed on 1 May 2021). Read counts per gene were generated by DEG-seq. DEGs in carcinoma or para-carcinoma tissues were defined as those differing by >2-fold between patients with

mild inflammation and fibrosis or moderate-to-severe inflammation and fibrosis, with an associated $p < 0.05$. Heat map and volcano plots were generated based on gene expression profiles and drawn using R package ggplot2 (version 3.3.5).

To predict the biological functions and signaling pathways of DEGs, we used clusterProfiler (version 4.2.2) in R to analyze their enrichment in Gene Ontology (GO) terms and Kyoto Encyclopedia for Genes and Genomes (KEGG) pathways. The threshold for enrichment was $p < 0.05$.

RNA-seq data were normalized, then entered into the CIBERSORT algorithm with 1000 iterations and the LM22 gene signature in order to quantify the relative proportions of 22 immune cell types [26].

### 2.3. Influence of Inflammation and Fibrosis on Cell Types Present in HCC Tumors and Para-Cancerous Tissues

#### 2.3.1. Sample Collection

Carcinoma and para-carcinoma tissues that had been surgically resected from 37 patients with non-cirrhotic HBV-associated HCC were analyzed using CyTOF. Patients underwent resection at our institution between 2018 and 2019, and they satisfied the same inclusion and exclusion criteria as the patients in Section 2.1.1; all but eight patients overlapped with those in Section 2.2.1. Inflammation and fibrosis were mild in 18 subjects and moderate-to-severe in 19.

After resection, tissue samples were placed immediately in pre-cooled Dulbecco's modified Eagle medium supplemented with 2% fetal bovine serum (FBS), 0.5% fluconazole antifungal solution, and 1% penicillin/streptomycin, then rapidly transported to the laboratory on ice. Tissues were dissociated into single-cell suspensions, red blood cells were removed, and the samples were stored in liquid nitrogen [27].

#### 2.3.2. Antibodies Labeled with Lanthanide Metal

Antibodies (Table 2) for labeling cells were purchased from Fluidigm (San Francisco, CA, USA) and Biolegend (San Diego, CA, USA). Antibodies for mass cytometry were conjugated to isotopes using a MaxPar X8 Antibody Labeling Kit (Fluidigm) according to the manufacturer's instructions. The polymer was mounted on the preset lanthanide metal of choice by co-incubation at 37 °C for 30 min. Then, the antibody (100 μg) was added to a 50 kDa ultrafiltration spin column containing R buffer (300 μL), then centrifuged at $12,000 \times g$ for 10 min at room temperature. The filtrate was discarded, and the concentrate was reduced in 4 mmol/L TCEP R-buffer bond breaker solution at 37 °C for 30 min. L-buffer (200 μL) and the polymerization mixture for lanthanide metal binding were mixed in a 3 kDa ultrafiltration spin column, then centrifuged at $12,000 \times g$ for 25 min at room temperature. The filtrate was discarded, and the concentrate was again mixed with L-buffer and centrifuged as above. The antibody was purified on a 50 kDa ultrafiltration spin column. After purification of polymer and antibodies, antibody conjugated with metal-loaded polymers was concentrated using a 50 kDa filter, incubated at 37 °C for 90 min, transferred to a new ultrafiltration tube, mixed with W-buffer (300 μL) for elution of the unbound polymer and metal, and centrifuged at $12,000 \times g$ for 10 min at room temperature. The filtrate was discarded. Then, the antibody with the metal label was recovered, transferred to a new microcentrifuge tube, and stored at 4 °C.

**Table 2.** CyTOF panel.

| Specificity | Antibody Clone | Metal lable | Source |
|---|---|---|---|
| CD19 | HIB19 | 142Nd | Fluidigm |
| CD20 | 2H7 | 161Dy | Biolegend |
| CD3 | UCHT1 | 154Sm | Fluidigm |
| CD4 | SK3 | 174Yb | Fluidigm |
| CD8a | RPA-T8 | 144Nd | Biolegend |
| CD11c | 3.9 | 146Nd | Fluidigm |
| CD14 | RMO52 | 148Nd | Fluidigm |
| CD25/IL-2R | 2A3 | 149Sm | Fluidigm |
| CD27 | LG.3A10 | 150Nd | Fluidigm |
| CD38 | HIT2 | 143Nd | Biolegend |
| CD45 | HI30 | 89Y | Fluidigm |
| CD45RA | HI100 | 170Er | Fluidigm |
| CD45RO | UCHL1 | 151Eu | Biolegend |
| CD66b | 80H3 | 162Dy | Fluidigm |
| CD86 | IT2.2 | 156Gd | Fluidigm |
| CD161 | HP-3G10 | 164Dy | Fluidigm |
| CD163 | GHI/61 | 145Nd | Fluidigm |
| CD196/CCR6 | G034E3 | 176Yb | Fluidigm |
| CD197/CCR7 | G043H7 | 167Er | Fluidigm |
| CD206/MMR | 15-2 | 168Er | Fluidigm |
| CD326/EpCAM | 9C4 | 141Pr | Fluidigm |
| HLA-DR | L243 | 173Yb | Fluidigm |
| CD274/PD-L1 | 29E.2A3 | 175Lu | Fluidigm |
| CD279/PD-1 | EH12.2H7 | 155Gd | Fluidigm |
| CD223/LAG-3 | 11C3C65 | 165Ho | Fluidigm |
| TIM-3 | F38-2E2 | 153Eu | Fluidigm |
| Foxp3 | 259D/C7 | 159Tb | Fluidigm |
| Granzyme B | GB11 | 171Yb | Fluidigm |
| IL-6 | MQ2-13A5 | 147Sm | Fluidigm |
| IL-10 | JES3-9D7 | 166Er | Fluidigm |
| IFN-γ | B27 | 158Gd | Fluidigm |
| TNF-a | Mab11 | 152Sm | Fluidigm |
| IL-17A | BL168 | 169Tm | Fluidigm |
| Ki-67 | B56 | 172Yb | Fluidigm |
| TGF-β | TW4-6H10 | 163Dy | Fluidigm |

2.3.3. Antibody Labeling of Single-Cell Suspensions

The single-cell suspensions from Section 2.3.1 were thawed. One million cells were taken from each sample and assessed for cell viability using trypan blue staining. Maxparr PBS (Fluidigm) was used to prepare cisplatin staining solution at a dilution of 1:10,000. The cisplatin solution (1 mL) was added to all cell suspensions to stain viable cells. The tubes were mixed well, incubated for 5 min at room temperature in the dark, then mixed with 2.5 volumes of cell staining buffer (CSB, Fluidigm). The mixture was centrifuged at $300\times g$ for 5 min at room temperature, and the supernatant was discarded. Cell suspensions were incubated with 50 μL of Fc receptor blocking solution (Biolegend) for 10 min, and incubated with surface antibody cocktail (50 μL) for 1 h at room temperature. The cells were then washed and incubated with 1 mL Maxpar® Nuclear Antigen Staining Buffer Set (Fluidigm) for 30 min at room temperature. The intracellular antibody cocktail was added to cell suspensions for 1 h. Samples were washed, then incubated with freshly prepared 1.6% paraformaldehyde (Thermo Fisher Scientific) at room temperature for 10 min. Finally, the samples were stained with Ir-Intercalator (Fluidigm) overnight at 4 °C. Cells were washed in CSB and deionized water to remove buffer salts, resuspended in Q™ Four Element Calibration Beads (Fluidigm), then analyzed on a Helios 2 CyTOF mass spectrometer-flow cytometer (Fluidigm).

2.3.4. Standardized Processing of Cytometric Data

The detection rate was set to < 500 cells/s. Data were imported into CyTOF software version 6.7 to merge the data files in a standardized manner to obtain flow cytometry standard data (FCS).

2.3.5. Processing and Analysis of Cytometric Data

Raw FCS files were normalized and imported into Cytobank (https://www.cytobank. org/, accessed on 14 February 2022) for gating. The default data transformation, hyperbolic arcsine, was used to exclude debris, dead cells, and doublets, and then the numbers of live cells were determined. To prevent larger samples from unduly influencing results, a maximum of 5000 cells were randomly selected for cluster analysis. Cells were visualized using plots of t-distributed stochastic neighbor embedding (t-SNE).

**3. Results**

*3.1. Influence of Inflammation and Fibrosis on Survival*

3.1.1. Patient Characteristics and Classification Based on Scheuer Score

Survival analysis was conducted on 293 patients with non-cirrhotic HBV-associated HCC (86.0% men), whose median age was 49 years (interquartile range 42–57), and mean tumor size at diagnosis was $7.8 \pm 4.3$ cm. Median follow-up was 40 months.

Inflammation and fibrosis were classified as mild in 189 patients, or as moderate-to-severe in the remaining 104 (Table 3). The two groups did not differ significantly in sex distribution, age, Barcelona Clinic liver cancer (BCLC) stage, serum albumin, aspartate transaminase (AST), alanine transaminase (ALT), total bilirubin, $\alpha$-fetoprotein (AFP), tumor number, tumor capsular, portal vein tumor thrombus, microvascular invasion, or satellite nodules. Tumors were significantly smaller and less differentiated in patients with moderate-to-severe inflammation and fibrosis, than in those with mild inflammation and fibrosis.

**Table 3.** Clinicopathological characteristics of patients with non-cirrhotic HBV-associated hepatocellular carcinoma who were treated with hepatectomy, stratified based on Scheuer severity of inflammation and fibrosis.

| Variable | | Before Propensity Score Matching | | | After Propensity Score Matching | | |
|---|---|---|---|---|---|---|---|
| | | Mild, $n = 189$ (%) | Moderate-to-Severe, $n = 104$ (%) | $p$ | Mild, $n = 67$ (%) | Moderate-to-Severe, $n = 67$ (%) | $p$ |
| Sex | | | | | | | |
| | Male | 159 (84.1) | 93 (89.4) | 0.291 | 62 (92.5) | 60 (89.6) | 0.545 |
| | Female | 30 (15.9) | 11 (10.6) | | 5 (7.5) | 7 (10.4) | |
| Age | | | | | | | |
| | <60 yr | 158 (83.6) | 81 (77.9) | 0.270 | 57 (85.1) | 53 (79.1) | 0.367 |
| | ≥60 yr | 31 (16.4) | 23 (22.1) | | 10 (14.9) | 14 (20.9) | |
| Tumor size | | | | | | | |
| | ≤5 cm | 60 (31.7) | 46 (44.2) | 0.042 * | 19 (28.4) | 18 (26.9) | 0.847 |
| | >5 cm | 129 (68.3) | 58 (55.3) | | 48 (71.6) | 49 (73.1) | |
| Number of tumors | | | | | | | |
| | <2 | 150 (79.4) | 79 (76.0) | 0.555 | 53 (79.1) | 54 (80.6) | 0.829 |
| | ≥2 | 39 (20.6) | 25 (24.0) | | 14 (20.9) | 13 (19.4) | |
| Tumor capsule | | | | | | | |
| | Complete | 140 (74.1) | 79 (76.0) | 0.780 | 50 (74.6) | 51 (76.1) | 0.841 |
| | Incomplete | 49 (25.9) | 25 (24.0) | | 17 (25.4) | 16 (23.9) | |
| MVI | | | | | | | |
| | Negative | 105 (55.6) | 53 (51.0) | 0.465 | 40 (59.7) | 36 (53.7) | 0.486 |
| | Positive | 84 (44.4) | 51 (49.0) | | 27 (40.3) | 31 (46.3) | |
| BCLC stage | | | | | | | |
| | A–B | 142 (75.1) | 71 (68.3) | 0.220 | 44 (65.7) | 42 (63.7) | 0.719 |
| | C | 47 (24.9) | 33 (31.7) | | 23 (34.3) | 25 (37.3) | |

**Table 3.** *Cont.*

| Variable | Before Propensity Score Matching | | | After Propensity Score Matching | | |
|---|---|---|---|---|---|---|
| | Mild, *n* = 189 (%) | Moderate-to-Severe, *n* = 104 (%) | *p* | Mild, *n* = 67 (%) | Moderate-to-Severe, *n* = 67 (%) | *p* |
| Edmondson grade | | | | | | |
| I–II | 103 (54.5) | 39 (37.5) | 0.007 * | 26 (38.9) | 28 (41.8) | 0.725 |
| III–IV | 86 (45.5) | 65 (63.5) | | 41 (61.2) | 39 (58.2) | |
| Serum albumin | | | | | | |
| <35 g/L | 17 (9.0) | 15 (14.4) | 0.173 | 6 (9.9) | 9 (13.4) | 0.411 |
| ≥35 g/L | 172 (91.0) | 89 (85.6) | | 61 (91.0) | 58 (86.6) | |
| ALT | | | | | | |
| ≤40 U/L | 114 (60.3) | 56 (53.8) | 0.323 | 40 (59.7) | 38 (56.7) | 0.726 |
| >40 U/L | 75 (39.7) | 48 (46.2) | | 27 (40.3) | 29 (43.3) | |
| AST | | | | | | |
| ≤40 U/L | 104 (55.0) | 45 (43.3) | 0.067 | 29 (43.3) | 29 (43.3) | 1.000 |
| >40 U/L | 85 (45.0) | 59 (56.7) | | 38 (56.7) | 38 (56.7) | |
| TBil | | | | | | |
| ≤17.1 μmol/mL | 161 (85.2) | 89 (85.6) | 1.000 | 57 (85.1) | 56 (83.6) | 0.812 |
| >17.1 μmol/mL | 28 (14.8) | 15 (14.4) | | 10 (14.9) | 11 (16.4) | |
| AFP | | | | | | |
| <400 ng/mL | 92 (48.7) | 56 (53.8) | 0.464 | 27 (40.3) | 38 (56.7) | 0.057 |
| ≥400 ng/mL | 97 (51.3) | 48 (46.2) | | 40 (59.7) | 29 (43.3) | |
| PV thrombosis | | | | | | |
| Absence | 153 (81.0) | 80 (76.9) | 0.451 | 49 (73.1) | 46 (68.7) | 0.568 |
| Presence | 36 (19.0) | 24 (23.1) | | 18 (26.9) | 21 (31.3) | |
| Satellite nodule | | | | | | |
| Absence | 169 (89.4) | 87 (83.7) | 0.198 | 59 (88.1) | 52 (77.6) | 0.109 |
| Presence | 20 (10.6) | 17 (16.3) | | 8 (11.9) | 15 (22.4) | |

Values are *n* (%), unless otherwise noted. Abbreviations: AFP, α-fetoprotein; ALT, alanine aminotransferase; AST, aspartate aminotransferase; BCLC, Barcelona Clinic Liver Cancer staging system; MVI, microvascular invasion; PV, portal vein; TBil, total bilirubin. *: <0.05.

Pairs of patients with mild or moderate-to-severe inflammation and fibrosis were matched 1:1 based on propensity scoring. The resulting 67 pairs showed no significant clinicopathological differences (Table 3), confirming the effectiveness of the matching procedure.

### 3.1.2. Prognostic Power of Inflammation and Fibrosis

Among all patients without propensity score matching, OS and RFS rates at 1, 2, and 5 years were significantly better among those with mild inflammation and fibrosis than among those with moderate-to-severe inflammation and fibrosis (Figure 1A,B). Univariate analysis revealed the following eight variables to be significant predictors of increased risk of mortality and recurrence: moderate-to-severe inflammation and fibrosis, tumor > 5 cm, multiple tumors, MVI, BCLC stage C, Edmondson grade III–IV, and presence of portal vein thrombosis or satellite nodules. Multivariate Cox analysis identified poor tumor differentiation as an independent predictor of OS (HR 2.044, 95% CI 1.270–3.291; *p* = 0.003) and RFS (HR 1.460, 95% CI 1.050–2.031; *p* = 0.024; Table 4). It identified the following factors as independent predictors of only RFS: moderate-to-severe inflammation and fibrosis (HR 1.439, 95% CI 1.028–2.015; *p* = 0.034), multiple tumors (HR 1.547, 95% CI 1.068–2.242; *p* = 0.021), BCLC stage C (HR 2.255, 95% CI 1.310–3.881; *p* = 0.003), and presence of satellite nodules (HR 1.706, 95% CI 1.091–2.668; *p* = 0.019).

Among patients matched based on propensity score, OS and RFS rates at 1, 2, and 5 years were significantly better among those with mild inflammation and fibrosis than among those with moderate-to-severe inflammation and fibrosis (Figure 1C,D). Univariate analysis revealed that the following four variables were significant predictors of both OS and RFS: moderate-to-severe inflammation and fibrosis, BCLC stage C, presence of

portal vein thrombosis, and MVI. Multivariate Cox analysis showed that moderate-to-severe inflammation and fibrosis was an independent predictor of OS (HR 2.091, 95% CI 1.118–3.911; *p* = 0.021) and RFS (HR 1.632, 95% CI 1.015–2.624; *p* = 0.043), BCLC stage C was an independent predictor of only RFS (HR 2.428, 95% CI 1.083–5.444; *p* = 0.031), and poor tumor differentiation was a predictor of only OS (HR 2.315, 95% CI 1.184–4.525; *p* = 0.014; Table 5).

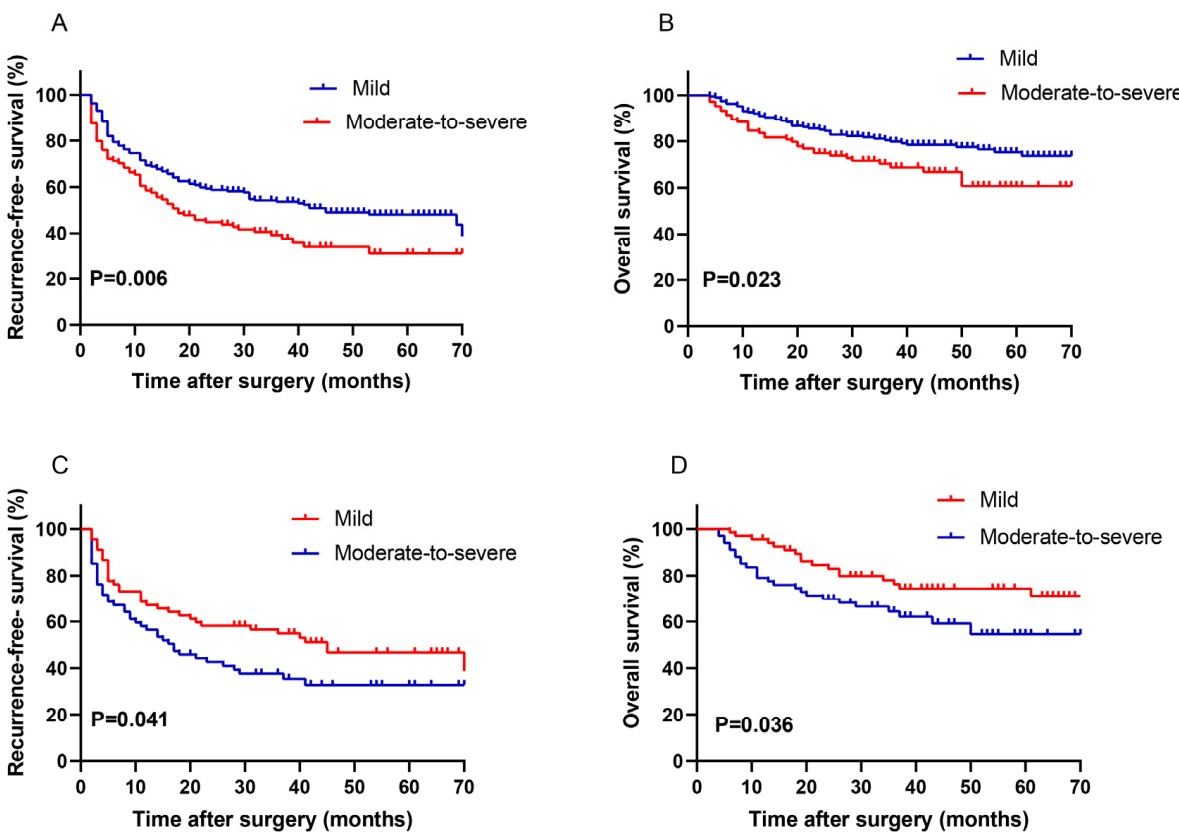

**Figure 1.** The prognostic significance of the inflammation and fibrosis. (**A**,**B**) Before propensity score match, recurrence-free survival (**A**) and overall survival (**B**) of non-cirrhotic HBV-associated HCC patients after hepatectomy. (**C**,**D**) After propensity score match, recurrence-free survival (**C**) and overall survival (**D**) of non-cirrhotic HBV-associated HCC patients after hepatectomy. Patients were stratified based on the Scheuer system.

**Table 4.** Uni- and multivariate analyses to identify predictors of recurrence-free and overall post-hepatectomy survival of patients with non-cirrhotic HBV-associated HCC, before propensity score matching.

| Variable | Overall Survival | | | | Recurrence-Free Survival | | | |
| --- | --- | --- | --- | --- | --- | --- | --- | --- |
| | Univariate HR (95% CI) | *p* | Multivariate HR (95% CI) | *p* | Univariate HR (95% CI) | *p* | Multivariate HR (95% CI) | *p* |
| Sex (male) | 1.661 (0.799–3.454) | 0.174 | | | 1.595 (0.978–2.609) | 0.061 | | |
| Age (≥60 yr) | 0.848 (0.473–1.521) | 0.581 | | | 1.117 (0.762–1.637) | 0.570 | | |
| Inflammation and fibrosis (Scheuer group) | 1.684 (1.047–2.598) | 0.031 * | 1.543 (0.971–2.451) | 0.066 | 1.530 (1.114–2.100) | 0.009 * | 1.439 (1.028–2.015) | 0.034 * |
| Tumor size (>5 cm) | 2.121 (1.265–3.558) | 0.004 * | 1.461 (0.833–2.562) | 0.186 | 1.862 (1.325–2.616) | <0.001 * | 1.403 (0.962–2.048) | 0.079 |
| Number of tumors (multiple) | 1.703 (1.045–2.775) | 0.033 * | 1.401 (0.833–2.562) | 0.186 | 1.942 (1.380–2.734) | <0.001 * | 1.547 (1.068–2.242) | 0.021 * |

**Table 4.** *Cont.*

| Variable | Overall Survival | | | | Recurrence-Free Survival | | | |
|---|---|---|---|---|---|---|---|---|
| | Univariate HR (95% CI) | p | Multivariate HR (95% CI) | p | Univariate HR (95% CI) | p | Multivariate HR (95% CI) | p |
| Tumor capsule (incomplete) | 1.389 (0.858–2.251) | 0.182 | | | 1.495 (1.067–2.095) | 0.019 * | 1.265 (0.875–1.830) | 0.212 |
| MVI (positive) | 2.339 (1.475–3.707) | <0.001 * | 1.606 (0.992–2.600) | 0.054 | 1.966 (1.437–2.688) | <0.001 * | 1.386 (0.987–1.947) | 0.060 |
| BCLC stage C | 3.393 (2.166–5.314) | <0.001 * | 1.560 (0.684–3.560) | 0.291 | 2.399 (1.738–3.312) | <0.001 * | 2.255 (1.310–3.881) | 0.003 * |
| Edmondson grade (III–IV) | 1.988 (1.274–3.102) | 0.002 * | 2.044 (1.270–3.291) | 0.003 * | 1.635 (1.204–2.220) | 0.002 * | 1.460 (1.050–2.031) | 0.024 * |
| Serum albumin (≥35 g/L) | 0.713 (0.366–1.390) | 0.321 | | | 0.823 (0.515–1.315) | 0.415 | | |
| ALT (>40 U/L) | 0.991 (0.629–1.561) | 0.969 | | | 1.031 (0.754–1.411) | 0.848 | | |
| AST (>40 U/L) | 1.502 (0.963–2.344) | 0.073 | | | 1.606 (1.177–2.191) | 0.003 * | 1.369 (0.980–1.912) | 0.066 |
| TBil (>17.1 μmol/mL) | 0.757 (0.733–1.517) | 0.432 | | | 0.859(0.543–1.360) | 0.517 | | |
| AFP (≥400 ng/mL) | 1.453 (0.928–2.277) | 0.103 | | | 1.515 (1.111–2.068) | 0.009 * | 1.155 (0.832–1.603) | 0.391 |
| PV thrombosis (presence) | 3.404 (2.156–5.373) | <0.001 * | 1.980 (0.858–4.569) | 0.110 | 1.855 (1.303–2.640) | 0.001 * | 0.644 (0.364–1.136) | 0.129 |
| Satellite nodule (presence) | 2.356 (1.374–4.041) | 0.002 * | 1.505 (0.840–2.698) | 0.169 | 2.413 (1.616–3.604) | <0.001 * | 1.706 (1.091–2.668) | 0.019 * |

Abbreviations: AFP, α-fetoprotein; ALT, alanine aminotransferase; AST, aspartate aminotransferase; BCLC, Barcelona Clinic Liver Cancer staging system; HR, hazard ratio; MVI, microvascular invasion; PV, portal vein; TBil, total bilirubin. *: <0.05.

**Table 5.** Uni- and multivariate analyses to identify predictors of recurrence-free and overall post-hepatectomy survival of patients with non-cirrhotic hepatis B-associated HCC, after propensity score matching.

| Variable | Overall Survival | | | | Recurrence-Free Survival | | | |
|---|---|---|---|---|---|---|---|---|
| | Univariate HR (95% CI) | p | Multivariate HR (95% CI) | p | Univariate HR (95% CI) | p | Multivariate HR (95% CI) | p |
| Sex (male) | 1.877 (0.569–3.759) | 0.301 | | | 1.577 (0.680–3.655) | 0.288 | | |
| Age (≥60 yr) | 1.270 (0.624–2.584) | 0.509 | | | 1.433 (0.836–2.457) | 0.191 | | |
| Inflammation and fibrosis (>7) | 1.907 (1.029–3.534) | 0.040 * | 2.091 (1.118–3.911) | 0.021 * | 1.576 (1.005–2.472) | 0.048 * | 1.632 (1.051–2.624) | 0.043 * |
| Tumor size (>5 cm) | 2.091 (1.000–4.374) | 0.050 * | 1.104 (0.496–2.457) | 0.809 | 1.737 (1.024–2.944) | 0.040 * | 0.936 (0.502–1.744) | 0.834 |
| Number of tumors (multiple) | 1.500 (0.756–2.978) | 0.247 | | | 1.925 (1.171–3.164) | 0.001 * | 1.568 (0.890–2.764) | 0.119 |
| Tumor capsule (incomplete) | 1.360 (0.709–2.608) | 0.355 | | | 1.180 (0.709–1.964) | 0.525 | | |
| MVI (positive) | 2.551 (1.385–4.699) | 0.003 | 1.861 (0.989–3.502) | 0.054 | 1.736 (1.110–2.714) | 0.016 * | 1.105 (0.681–1.795) | 0.685 |
| BCLC stage C | 3.780 (2.041–7.001) | <0.001 * | 2.144 (0.703–6.534) | 0.180 | 2.025 (1.290–3.177) | 0.002 * | 2.428 (1.083–5.444) | 0.031 * |
| Edmondson grade (III–IV) | 1.920 (1.002–3.680) | 0.049 * | 2.315 (1.184–4.525) | 0.014 * | 1.560 (0.978–2.487) | 0.062 | 1.517 (0.938–2.453) | 0.089 |
| Serum albumin (≥35 g/L) | 0.654 (0.276–1.551) | 0.335 | | | 0.912 (0.455–1.828) | 0.795 | | |
| ALT (>40 U/L) | 1.169 (0.643–2.125) | 0.608 | | | 1.060 (0.676–1.662) | 0.801 | | |
| AST (>40 U/L) | 1.614 (0.875–2.975) | 0.125 | | | 1.698 (1.069–2.697) | 0.025 * | 1.657 (0.990–2.774) | 0.055 |
| TBil (>17.1 μmol/mL) | 1.088 (0.484–2.445) | 0.839 | | | 0.876 (0.463–1.659) | 0.685 | | |

**Table 5.** *Cont.*

| Variable | Overall Survival | | | | Recurrence-Free Survival | | | |
| --- | --- | --- | --- | --- | --- | --- | --- | --- |
| | Univariate HR (95% CI) | *p* | Multivariate HR (95% CI) | *p* | Univariate HR (95% CI) | *p* | Multivariate HR (95% CI) | *p* |
| AFP (≥400 ng/mL) | 1.650 (0.895–3.044) | 0.109 | | | 1.468 (0.937–2.299) | 0.094 | | |
| PV thrombosis (presence) | 3.393 (1.861–6.186) | <0.001 * | 1.854 (0.636–5.350) | 0.259 | 1.691 (1.058–2.701) | 0.028 * | 0.729 (0.324–1.643) | 0.446 |
| Satellite nodule (presence) | 1.734 (0.853–3.523) | 0.128 | | | 2.244 (1.329–3.791) | 0.003 * | 1.785 (0.966–3.299) | 0.064 |

Abbreviations: AFP, α-fetoprotein; ALT, alanine aminotransferase; AST, aspartate aminotransferase; BCLC, Barcelona Clinic Liver Cancer staging system; HR, hazard ratio; MVI, microvascular invasion; PV, portal vein; TBil, total bilirubin. *: <0.05.

### *3.2. Influence of Inflammation and Fibrosis on the HCC Transcriptome*

#### 3.2.1. Analysis of DEGs

In analysis of carcinoma tissues, we identified 798 DEGs, among which 443 were upregulated and 355 downregulated among patients with moderate-to-severe inflammation and fibrosis compared to those with mild inflammation and fibrosis (Figure 2A,C). In analysis of para-carcinoma tissues, 315 significant DEGs were identified, of which 126 were upregulated and 189 were downregulated among patients with moderate-to-severe inflammation and fibrosis (Figure 2B,D).

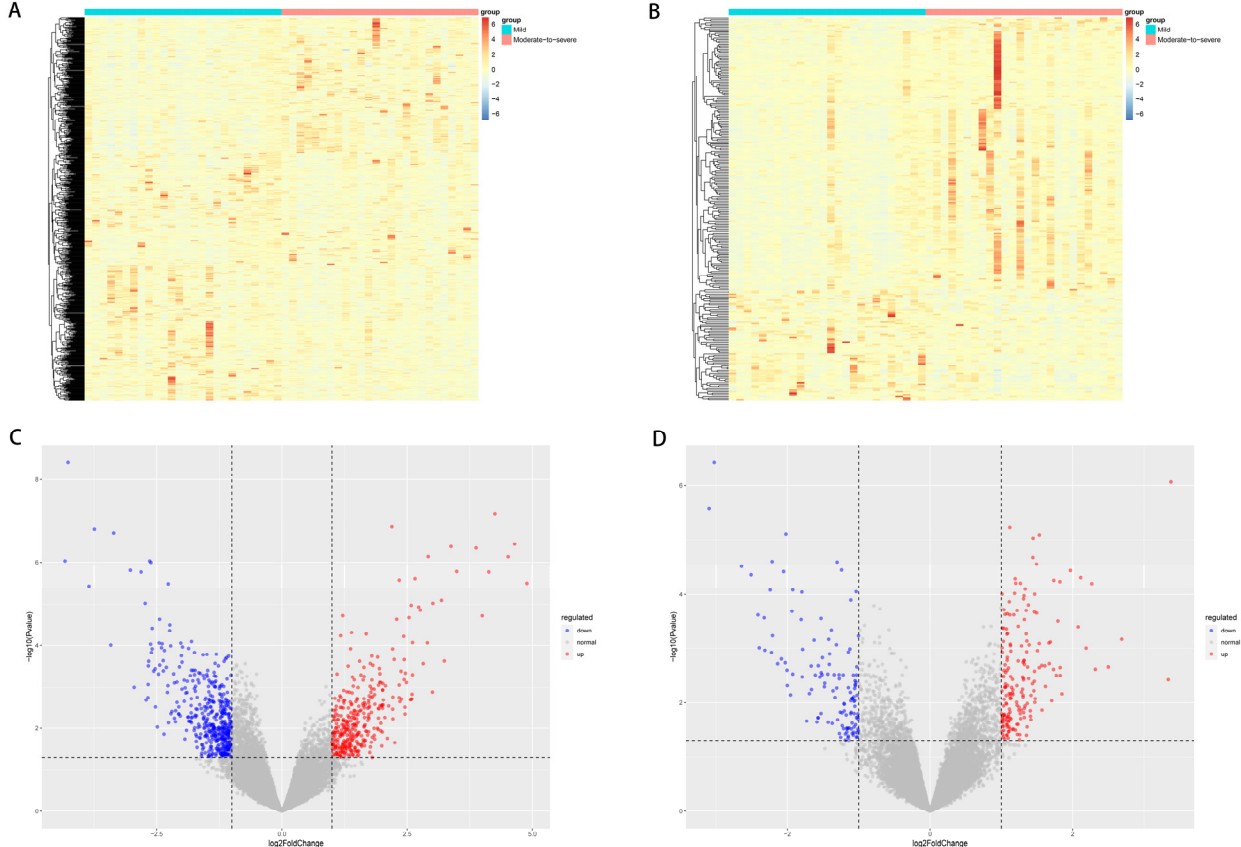

**Figure 2.** (**A,B**) Heat maps showing the clustering of genes differentially expressed between patients with moderate-to-severe or mild inflammation or fibrosis in (**A**) carcinoma tissues or (**B**) para-carcinoma tissues. (**C,D**) Volcano plots of genes differentially expressed between patients with moderate-to-severe or mild inflammation or fibrosis in (**C**) carcinoma tissues or (**D**) para-carcinoma tissues.

### 3.2.2. Analysis of GO Enrichment

DEGs upregulated in tumor tissue among patients with moderate-to-severe inflammation and fibrosis were enriched mainly in 28 biological processes such as regulation of DNA-templated transcription, (GO: 0006355), transcription from RNA polymerase II promoters (GO: 0006366), positive regulation of transcription (GO: 0045944), and positive regulation of cell proliferation (GO: 0008284) (Figure 3A). Downregulated DEGs were enriched mainly in 15 biological processes such as positive regulation of cytosolic calcium ion (GO: 0007204), cellular oxidant detoxification (GO: 0098869), and xenobiotic metabolic processes (GO: 0006805) (Figure 3B).

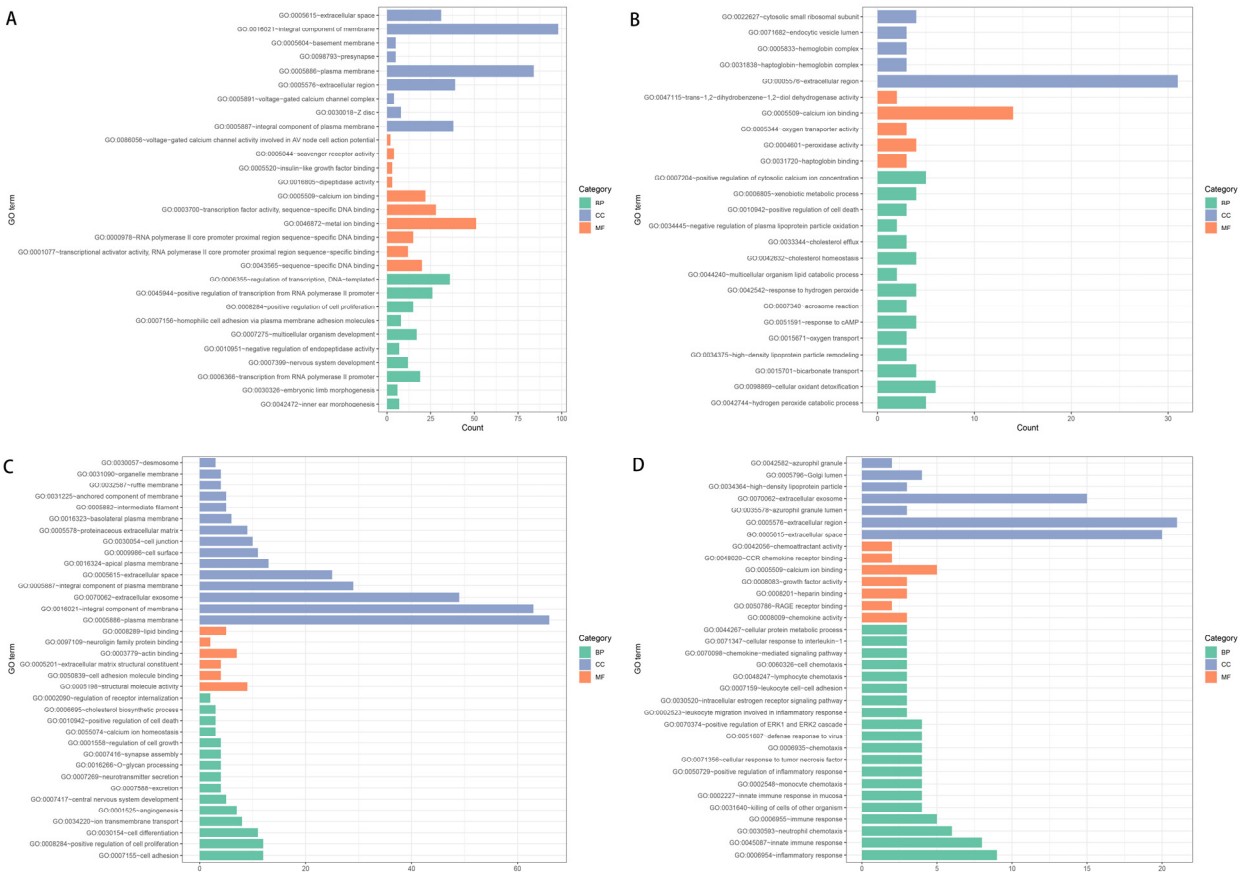

**Figure 3.** (**A**,**B**) Enrichment in Gene Ontology (GO) terms among genes in carcinoma tissues that were (**A**) upregulated or (**B**) downregulated among patients with moderate-to-severe inflammation and fibrosis relative to patients with mild inflammation and fibrosis. (**C**,**D**) Enrichment in GO terms among genes in para-carcinoma tissues that were (**C**) upregulated or (**D**) downregulated among patients with moderate-to-severe inflammation and fibrosis relative to patients with mild inflammation and fibrosis.

Upregulated DEGs were enriched mainly in 10 molecular functions such as calcium ion binding (GO: 0005509), proximal RNA polymerase II core promoters (GO: 0000978), and voltage-gated calcium channel activity involved in AV node cell action potential (GO: 0086056). Downregulated DEGs were enriched mainly in molecular functions such as peroxidase activity (GO: 0004601), haptoglobin binding (GO: 0031720), oxygen transporter activity (GO: 0005344), and trans-1,2-dihydrobenzene-1,2-diol (GO: 0047115).

Upregulated DEGs were enriched mainly in eight cellular components such as plasma membrane (GO: 0005886), extracellular regions (GO: 0005576), and extracellular spaces (GO: 0005615). Downregulated DEGs were enriched mainly in cellular components such as

cytosolic small ribosomal subunit (GO: 0022627), haptoglobin-hemoglobin complex (GO: 0031838), hemoglobin complex (GO: 0005833), and endocytic vesicle lumen (GO: 0071682).

GO enrichment among DEGs in para-cancerous tissue was substantially different (Figure 3C,D). DEGs upregulated in the para-tumoral tissue of patients with moderate-to-severe inflammation and fibrosis were enriched mainly in 14 biological processes such as positive regulation of cell proliferation (GO: 0008284), regulation of cell growth (GO: 0001558), cell differentiation (GO: 0030154), cell adhesion (GO: 0007155), and angiogenesis (GO: 0001525). Downregulated DEGs were enriched mainly in 33 biological processes such as innate immune response (GO: 0045087), innate immune response in mucosa (GO: 0002227), neutrophil chemotaxis (GO: 0030593), and monocyte chemotaxis (GO: 0002548).

The upregulated DEGs in para-tumoral tissue were enriched mainly in six molecular functions such as cell adhesion molecule binding (GO: 0050839), extracellular matrix structural constituents (GO: 0005201), and structural molecule activity (GO: 0005198). Downregulated DEGs were enriched mainly in seven molecular functions such as chemokine activity (GO: 0008009), growth factor activity (GO: 0008083), and chemoattractant activity (GO: 0042056).

Upregulated DEGs in para-tumoral tissue were enriched mainly in fourteen cellular components such as plasma membrane (GO: 0005886), proteinaceous extracellular matrix (GO: 0005578), and extracellular space (GO: 0005615). Downregulated DEGs were enriched mainly in six cellular components such as the extracellular space (GO: 0005615) and extracellular region (GO: 0005576).

### 3.2.3. Analysis of KEGG Enrichment

DEGs upregulated in tumor tissue of patients with moderate-to-severe inflammation and fibrosis were involved mainly in PI3K-Akt signaling (hsa04151), Rap1 signaling (hsa04015), and MAPK signaling (hsa04010) (Figure 4A,B). In contrast, DEGs upregulated in para-carcinoma tissues of patients with moderate-to-severe inflammation and fibrosis were enriched in cell adhesion molecules (hsa04514), calcium signaling (hsa04020), IL-17 signaling (hsa04657), and Hedgehog signaling (hsa04340), which were activated. However, they were enriched in p53 signaling, which was inhibited (Figure 4C,D).

### 3.2.4. CIBERSORT Analysis

Based on bulk RNA sequencing, we identified differences in infiltration of para-carcinoma tissues by immunocytes between patients with moderate-to-severe or mild inflammation and fibrosis (Figure 5C,D). In the para-carcinoma tissues, moderate-to-severe inflammation and fibrosis were associated with significantly lower proportions of activated mast cells and monocytes, but a significantly higher proportion of regulatory T cells (Figure 5E). In contrast, we did not identify significant differences in the infiltration of tumor tissues by immunocytes between the two groups of patients (Figure 5A,B).

### 3.3. Influence of Inflammation and Fibrosis on Cell Types Present in HCC Tumors and Para-Cancerous Tissues

Analysis of tumor tissues detected 21 clusters of CD45+ cells, including 7 clusters of CD4+ T cells (CD3+, CD4+; clusters 7, 8, 10, 12, 14, 15, 21), 5 clusters of CD8+ cells (CD3+, CD8+; clusters 13, 16, 17, 19, 20), 2 clusters of B cells (CD19+, CD20+; clusters 1 and 4), 1 cluster of tumor-associated macrophages (CD3-, CD14+, HLA-DR, CD163+, CD206+; cluster 2), 1 cluster of tumor-associated neutrophils (CD3-, CD66b+; cluster 5) and 5 cell clusters that could not be fully classified (clusters 3, 6, 9, 11, 18) (Figure 6A–D).

Clusters 10 and 12 contained regulatory T cells (CD3+, CD4+, CD8-, CD25+, FOXP3+). Cluster 8 contained PD-1+ CD4+ T cells (CD3+, CD4+, CD279+), while cluster 19 was PD-1+ CD8+ T cells (CD3+, CD8+, CD279+). Cluster 2 contained tumor-associated macrophages with an M2 phenotype.

Cluster 7, which contained the Th17 subset of CD4+ T cells (CD3+, CD4+, CD196+, HLA-DR), was significantly more abundant in tumor tissues of patients with moderate-to-

severe inflammation and fibrosis than in tumor tissues of patients with mild inflammation and fibrosis (Figure 7A). In addition, the immune exhaustion marker PD-L1 (CD274) was upregulated in CD8+ T cells from patients with moderate-to-severe inflammation and fibrosis compared to its expression in patients with mild inflammation and fibrosis; although, expression was similar for other markers of immune exhaustion (TIM3, LAG-3, PD-1) (Figure 7B).

The clustering of CD45+ cells in para-cancerous tissue was different from that in tumor tissues. Our analysis detected 19 cell clusters in para-cancerous tissue, including 2 clusters of CD4+ T cells (CD3+, CD4+; clusters 3, 19), 10 clusters of CD8+ cells (CD3+, CD8+; clusters 1, 2, 4, 5, 6, 8, 9, 11, 13, 17), 1 cluster of B cells (CD19+, CD20+; cluster 14), 1 cluster of tumor-associated macrophages (CD3-, CD14+, HLA-DR, CD163+, CD206+; cluster 10), 1 cluster of tumor-associated neutrophils (CD3-,CD66b+; cluster 16), and 4 cell clusters that could not be fully classified (clusters 7, 12, 15, 18; Figure 8A–D). We did not detect significant differences in the proportions of cell clusters or in the expression of immune exhaustion markers (TIM3, LAG-3, PD-1, PD-L1) between patients with moderate-to-severe or mild inflammation and fibrosis.

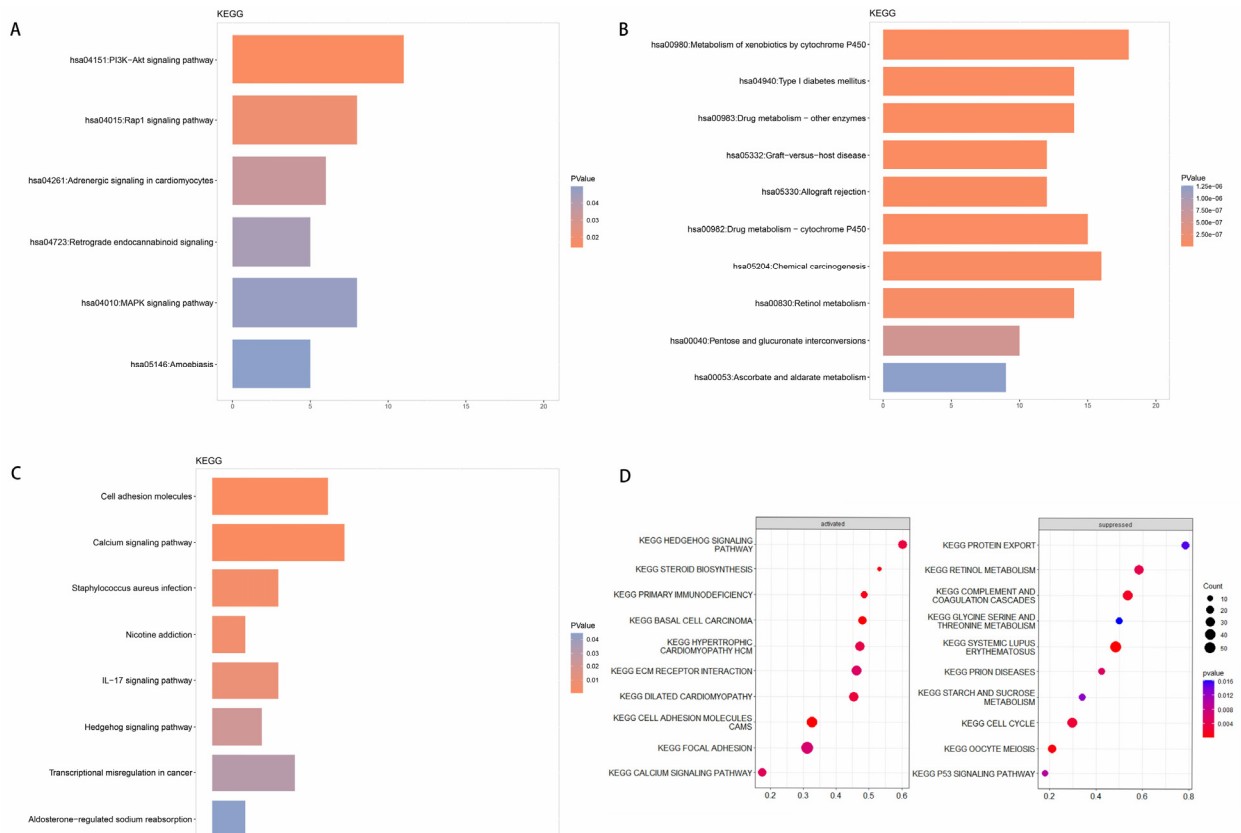

**Figure 4.** (**A**–**D**) Enrichment in Kyoto Encyclopedia of Genes and Genomes (KEGG) pathways among genes in carcinoma tissues that were (**A**) upregulated or (**B**) downregulated among patients with moderate-to-severe inflammation and fibrosis relative to patients with mild inflammation and fibrosis. (**C**) Enrichment in KEGG pathways among genes in para-carcinoma tissues in patients with moderate-to-severe inflammation and fibrosis relative to patients with mild inflammation and fibrosis. (**D**) State of activation or inhibition of enriched KEGG pathways.

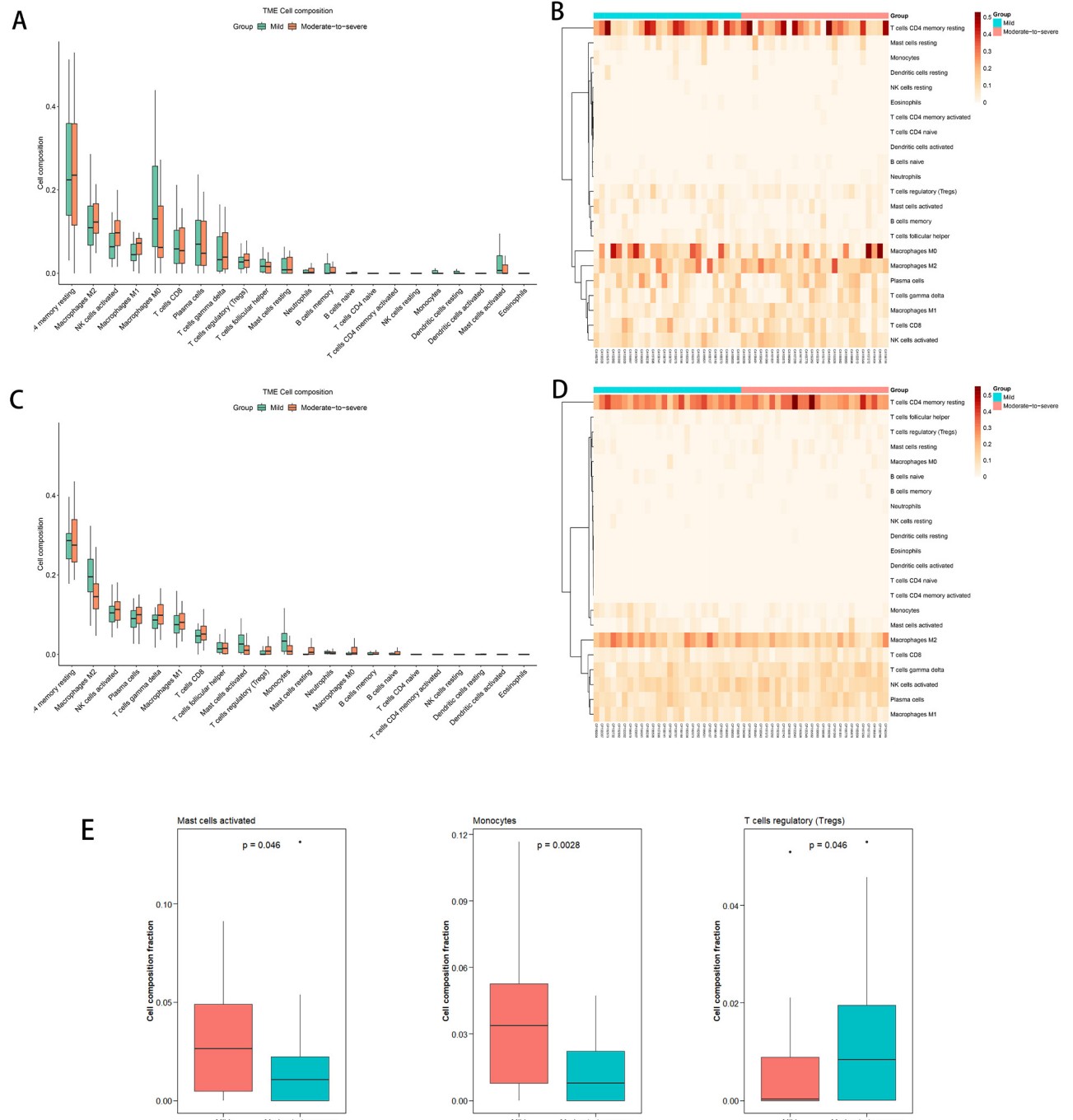

**Figure 5.** CIBRSORT comparing immune cell populations between patients with moderate-to-severe or mild inflammation and fibrosis based on analysis of (**A**,**B**) tumor tissue or (**C**,**D**) para-carcinoma tissue. (**E**) CIBERSORT comparing levels of mast cells activated, monocytes and regulatory T cells in para-carcinoma tissues between patients with moderate-to-severe, or mild inflammation and fibrosis.

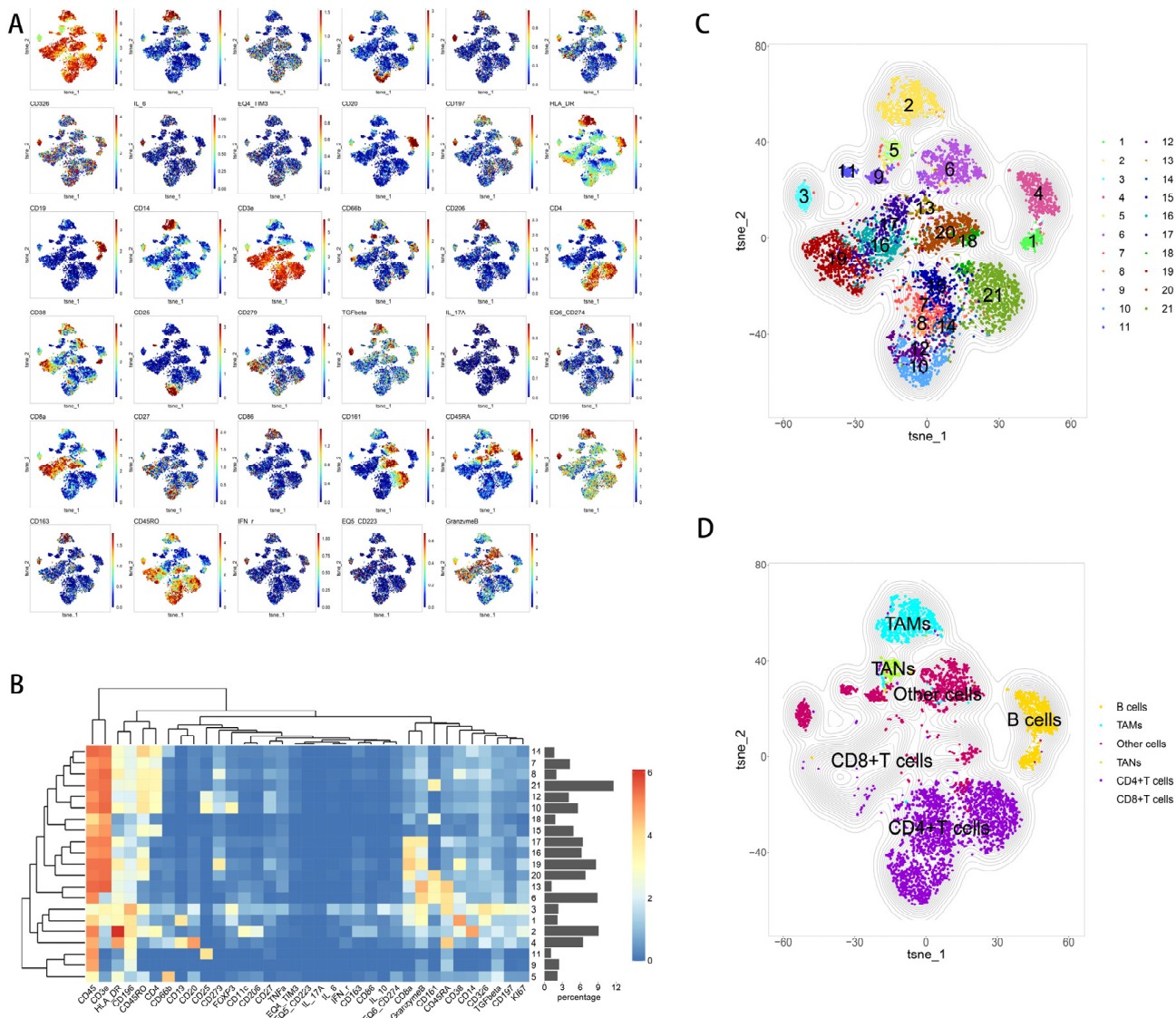

**Figure 6.** Expression of immune marker genes in 21 subsets of immune cells in tumors. (**A**) t-SNE diagram of the expression of each immune marker. (**B**) Heatmap of the expression of all immune-related markers after normalization. (**C,D**) t-SNE plots of immune cells that infiltrate tumor tissues.

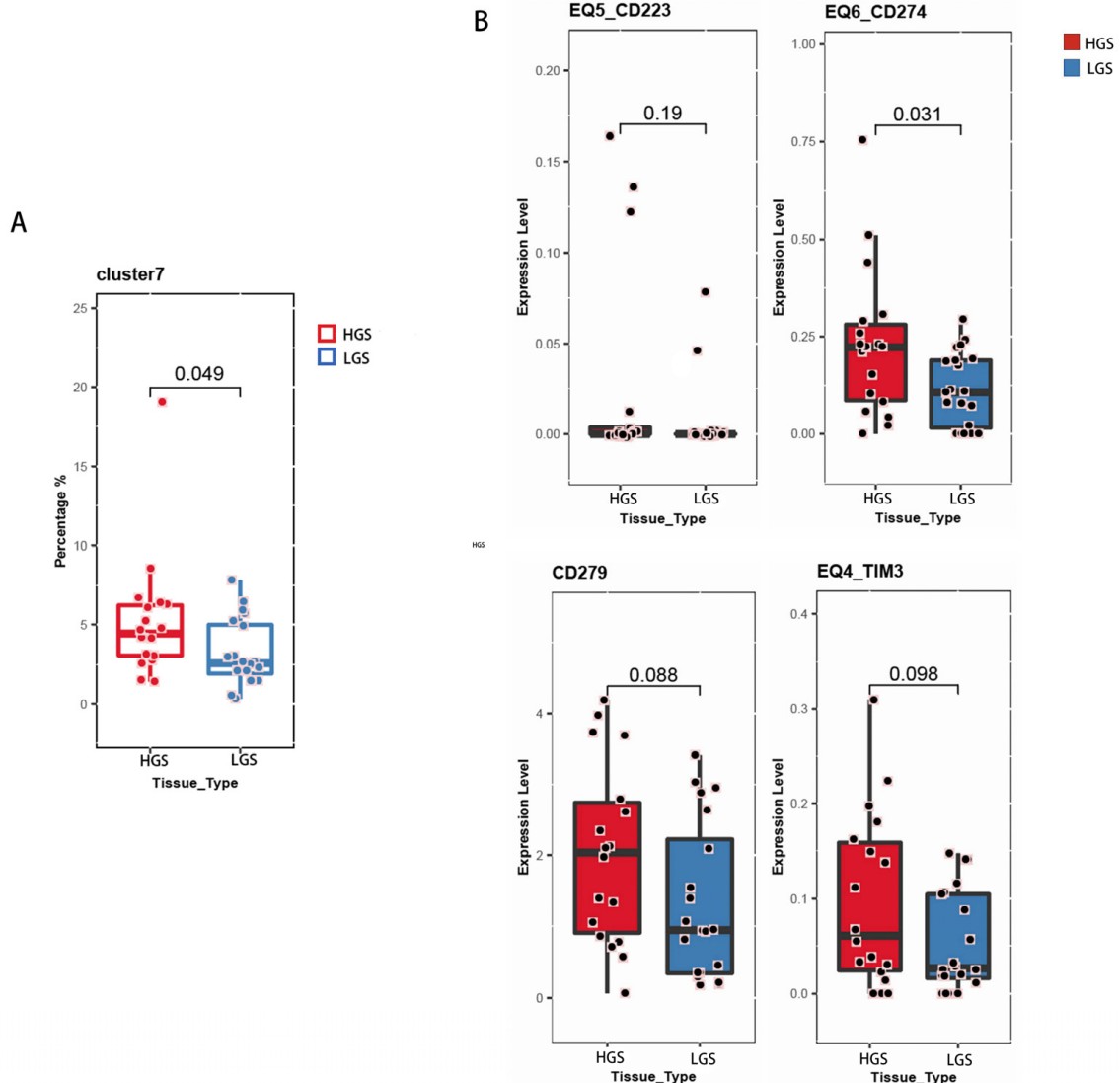

**Figure 7.** (**A**) Box plots of proportions of all CD4+ T cells that were Th17 cells in carcinoma tissues from patients with moderate-to-severe or mild inflammation and fibrosis. (**B**) Box plots showing expression of PD-L1 in CD8+ T cells in carcinoma tissues in patients with moderate-to-severe or mild inflammation and fibrosis.

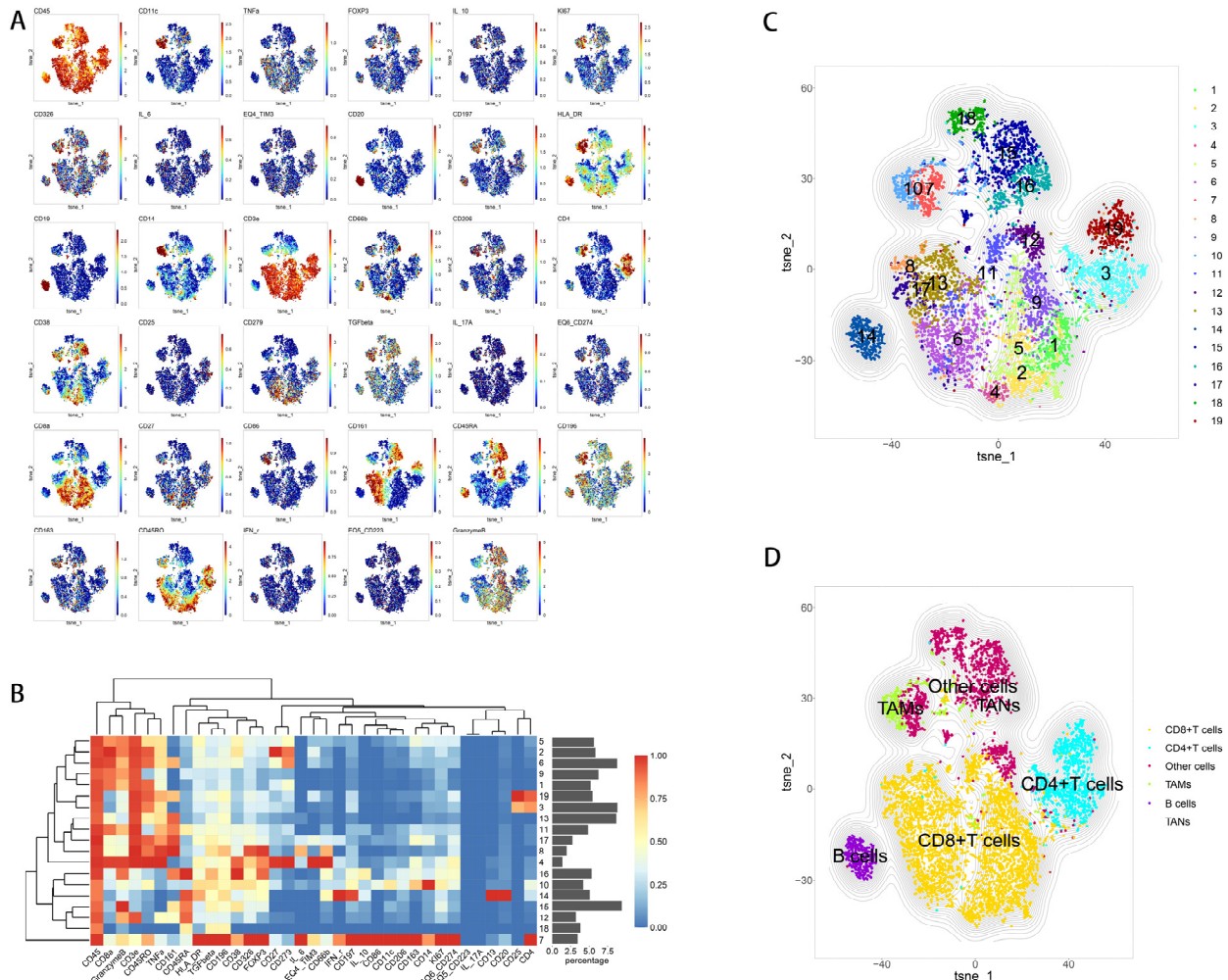

**Figure 8.** (**A**) t-SNE diagram of the expression of each immune marker. (**B**) Heatmap of the expression of all immune-related markers after normalization. (**C,D**) t-SNE plots of immune cells that infiltrate tumor tissues.

## 4. Discussion

In this study, we analyzed the effects of inflammation and fibrosis on the survival and transcriptome of patients with HBV-associated HCC who underwent hepatic resection, and whose para-carcinoma tissues did not meet the diagnostic criteria for cirrhosis. In our sample, univariate and multivariate regression identified moderate-to-severe inflammation and fibrosis as an independent risk factor for shorter RFS and OS. At the same time, our transcriptomic analysis linked moderate-to-severe inflammation and fibrosis to an increase in gene transcription, as well as signaling via PI3K-Akt, MAPK, and Ras in tumor tissues, which may drive tumor cell proliferation, invasion, and metastasis. Worse inflammation and fibrosis were also linked to an immunosuppressed tumor microenvironment, which in turn may facilitate tumor growth and spread.

Hepatis virus C (HCV) is one of the important causes of viral hepatitis, and chronic HCV infection displayed a high rate of progression to liver cirrhosis over a long-term follow-up compared with HBV infection [28]. However, China is a region with a high prevalence of HBV infection, so this study focuses on HBV-associated HCC.

Our results are consistent with the idea that inflammation drives HCC and many other cancers [29]. Inflammation can induce the production of chemokines and cytokines, which can alter cell proliferation and apoptosis, immune responses, and growth of new blood vessels [30,31], ultimately promoting tumor onset, development and recurrence.

The presence and extent of fibrosis after HBV infection marks the severity and duration of the disease [16]. As fibrosis progresses to cirrhosis, other events that are positively associated with fibrosis, such as chronic inflammation, hepatocellular injury, regeneration, etc., are also present for an equally long period of time and act synergistically, which becomes a driving factor in the development of HCC. In HCC patients with cirrhosis, although the primary tumor was surgically removed, it did not change the background of cirrhosis in their liver, and their para-carcinoma tissues still had drivers that induced HCC formation. These issues explain the prevalence of poorer prognosis for HCC patients with cirrhosis than those without cirrhosis.

Other studies, like the present one, have concluded that the degree of liver fibrosis is an independent prognostic factor for patients with hepatis virus-associated HCC [32,33], though some work calls this idea into question [13,34]. This discrepancy likely reflects differences in patient samples and study methods. If our results can be replicated in other populations, they suggest that people with moderate-to-severe inflammation and fibrosis of liver cancer with chronic HBV infection could benefit from antiviral and anti-fibrosis therapies to prevent progression to liver cancer. Such therapies have already been shown to improve prognosis of patients with HBV-associated HCC [35–38].

Patients with moderate-to-severe inflammation and fibrosis may also benefit from adjuvant treatment. Various studies are underway to explore the benefits of antiviral therapy with nucleoside/nucleotide analogues [21], anti-fibrosis therapy, and traditional Chinese medicine [39] for patients with HBV-associated HCC. Accurate assessment of liver inflammation and fibrosis may help optimize the type, dose, and timing of such treatments to prevent recurrence.

In our sample, AFP was not an independent prognostic factor for OS or RFS. This contrasts with several studies linking higher AFP in HCC patients to more aggressive tumors, which in turn are associated with greater recurrence and worse survival [40–42]. The reason for this discrepancy may be that our patients generally had milder liver injury and were in earlier stages of HCC than the patients in those previous studies, since we excluded those who were ineligible for radical resection and those with cirrhosis.

Many studies have shown MVI to predict RFS and OS in HCC patients after surgical resection [43,44]. In our sample, MVI approached, but did not achieve, statistical significance as an independent risk factor for OS and RFS. We still consider it to be clinically important in our daily practice, so further studies should continue to explore its prognostic significance.

Based on RNA sequencing, we found that signaling pathways involving PI3K-Akt, MAPK, and Rap1 were activated in the tumors of patients with moderate-to-severe inflammation and fibrosis, relative to the tumors of patients with mild inflammation and fibrosis. These pathways not only promote tumor progression, but they interact with each other to promote tumor invasion and metastasis [45–48]. Worse inflammation and fibrosis in our sample was associated with activation of growth and adhesion pathways, and inhibition of tumor-inhibiting pathways, consistent with previous studies [49–55]. Additionally, within HCC tumors, we found a higher proportion of Th17 cells among patients with moderate-to-severe inflammation and fibrosis, and a larger number of Th17 cells correlates with tumor growth and progression [56–59]. In para-carcinoma tissues, we found that the proportions of activated mast cells and monocytes were significantly lower among patients with moderate-to-severe inflammation and fibrosis than among patients with mild inflammation and fibrosis, suggesting weakened anti-tumor immune responses [60,61]. And the high proportion of regulatory T cells suggests an immunosuppressive tumor microenvironment [62–64]. These findings suggest that worse inflammation and fibrosis in our patients was associated with greater malignancy, which could help explain their worse prognosis.

Consistent with this idea, we found that CD8+ T cells in carcinoma tissues expressed higher levels of PD-L1 in patients with moderate-to-severe inflammation and fibrosis than in patients with mild inflammation and fibrosis. Higher levels of PD-L1 inhibit T cell

migration and proliferation, induce T cell apoptosis, and help tumor cells resist apoptosis induced by immune cells, all of which promote tumor progression [65–67] and worsen prognosis [68,69]. An immunosuppressed tumor microenvironment makes it easier for tumor cells to escape from surveillance and metastasize. These considerations imply that monoclonal antibody against PD-L1 may be effective immunotherapy for HCC patients with moderate-to-severe inflammation and fibrosis.

RNA sequencing and time-of-flight cytometry of para-cancerous tissue suggested up-regulation of cell adhesion, proliferation, differentiation, extracellular matrix formation, cell growth, and angiogenesis in patients with moderate-to-severe inflammation and fibrosis, relative to the corresponding tissues from patients with mild inflammation and fibrosis. This suggests that greater inflammation and fibrosis correlate with a para-tumor microenvironment more conducive to tumor cell colonization and growth. At the same time, our analysis suggests that such a microenvironment involves weakened innate and mucosal immune responses, and chemotaxis of neutrophils and monocytes. These findings may also help to explain the worse prognosis of patients with moderate-to-severe inflammation and fibrosis.

Our study presents several limitations. Our sample was small and came from a single center, so our findings should be verified and extended in larger, preferably prospective and multicenter studies. We did not consider how genetic differences may influence patient survival or the transcriptome in tumor and para-cancerous tissues. Genotype should be considered when individualizing HCC treatment. It is thus of interest to include genotype-based stratification of non-cirrhotic HCC patients.

Despite these limitations, our study provides evidence that the combination of inflammation and fibrosis can effectively predict the prognosis of patients with HBV-associated HCC without cirrhosis. Reducing liver inflammation and fibrosis may prevent recurrence and improve survival of such patients after hepatectomy.

## 5. Conclusions

Worse inflammation and fibrosis in non-cirrhotic HBV-associated HCC is associated with worse prognosis, which may reflect more aggressive tumor behavior and an immuno-suppressed, pro-metastatic tumor microenvironment.

**Author Contributions:** All authors contributed to the study conception and design. Conceptualization, Y.L., J.-F.Z. and B.-D.X.; data curation, Y.L., J.-F.Z., J.Z., G.-H.Z., Y.-K.L., J.-T.H. and B.-D.X.; formal analysis, Y.L., J.-F.Z., J.Z., G.-H.Z., Y.-K.L. and J.-T.H.; investigation, Y.-K.L., J.-T.H. and X.H.; methodology, Y.L., J.-F.Z., J.Z. and G.-H.Z.; project administration, B.-D.X.; supervision, B.-D.X.; writing—original draft, Y.L.; writing—review and editing, Y.L. and B.-D.X. All authors have read and agreed to the published version of the manuscript.

**Funding:** This work was supported by grants from the National Natural Science Foundation of China (81960450), high-level innovation team and outstanding scholar program in Guangxi Colleges and Universities, "139" projects for training of high-level medical science talents from Guangxi, the Key Research and Development Project of Guangxi (AB20297009), the Key Laboratory of Early Prevention and Treatment for Regional High Frequency Tumor, Ministry of Education Guangxi, the Independent Research Project (GKE2019-ZZ07, GKE-ZZ202005, GKE-ZZ202111), Development and application of medical and health appropriate technology in Guangxi (S2019039).

**Institutional Review Board Statement:** This study was performed in line with the principles of the Declaration of Helsinki. Approval was granted by the Ethics Committee of Guangxi Medical University Cancer Hospital (Data: 11 November 2021/ Number: LW2021096).

**Informed Consent Statement:** This study is a retrospective study. Only the clinical data of patients are collected, and there is no intervention in the treatment plan of patients, which will not bring physiological risks to patients. The researchers will do their best to protect the information provided by patients who have disclosed personal private information, and have signed the relevant informed consent at the time of admission.

**Data Availability Statement:** The datasets generated during, or analyzed during, the current study are available from the corresponding author on reasonable request.

**Acknowledgments:** We would like to thank the patients for participating in this study, and the multidisciplinary team at the hospital. We thank A. Chapin Rodríguez for his language editing, which substantially improved the quality of the manuscript.

**Conflicts of Interest:** No potential conflict of interest relevant to this article were reported.

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
