# Peer review of "Inflammation and Fibrosis in Patients with Non-Cirrhotic Hepatitis B Virus-Associated Hepatocellular Carcinoma: Impact on Prognosis after Hepatectomy and Mechanisms Involved"

_curroncol, doi:10.3390/curroncol30010016_

Round 1
Reviewer 1 Report
In this study, Bang-De Xiang et al tried to investigate if the degree of inflammation and fibrosis (Based on the Scheuer score system) in para-carcinoma tissue can predict the prognosis of patients with hepatitis B virus (HBV)-associated hepatocellular carcinoma (HCC) after hepatectomy and explored the mechanisms through which inflammation and fibrosis might affect prognosis.
They considered the clinicopathological data from 293 patients with non-cirrhotic HBV-associated HCC who were treated by curative resection from 2012 to 2017. Rates of overall and recurrence-free survival were compared between the groups using Kaplan–Meier curves, and survival predictors were identified using Cox regression.
They found that patients with mild inflammation and fibrosis showed significantly better overall and recurrence-free survival than those with moderate-to-severe inflammation and fibrosis. Multivariate Cox regression confirmed that moderate-to-severe inflammation and fibrosis were independent risk factors for worse survival. RNA sequencing and CyTOF showed that more severe inflammation and fibrosis were associated with stronger invasion and migration by hepatocytes. In cancerous tissues, the proportion of Th17 cells promoting tumor progression was increased and CD8+ T cells expressed higher levels of PD-L1. They concluded that worse inflammation and fibrosis in non-cirrhotic HBV-associated HCC is associated with a worse prognosis, which may reflect more aggressive tumor behavior and an immunosuppressed, pro-metastatic tumor microenvironment.
The study is of clinical impact. However, to further support the impact of inflammation and fibrosis, the authors should specify whether other potential etiologies of liver damage were excluded. In this regard, the authors should recall the relevant role of alcohol intake in HBV patients as a significant factor for HCC development in HBV chronic liver disease as previously demonstrated (Natural course of chronic HCV and HBV infection and role of alcohol in the general population: the Dionysos Study. Am J Gastroenterol. 2008 Sep;103(9):2248-53.)
-Clusters 10 and 12 contained regulatory T cells (CD3+, CD4+, CD8-, CD25+, FOXP3+): this is an interesting finding worth mentioning in previous literature data. Recent literature data support the immunosuppressive role of CD4+ Foxp3 regulatory T cells in the immune microenvironment of HCC that develop in HBV-related liver disease, as recently reported (Hepatocellular carcinoma in viral and autoimmune liver diseases: Role of CD4+ CD25+ Foxp3+ regulatory T cells in the immune microenvironment. World J Gastroenterol. 2021 Jun 14;27(22):2994-3009.).
Author Response
Reply: We appreciate the reviewer’s suggestions and positive evaluation of our work. In the revised manuscript, we have properly supplemented the suggestions put forward by the reviewer. Meanwhile, we found that the two studies provided by the reviewer are very interesting and coincide with our research results. Therefore, we have quoted these two literatures. The details of the revision can be found in the revised manuscript (lines 53-56 and lines 506-507).
Reviewer 2 Report
1, All patients received antivirus therapy peri-operative....., Please make sure it was pre-op. peri-op, or post-op? and how about the HBV eradication rate in your patients?
2. in the 2nd paragraph of "introduction; the cirrhosis and prognosis after hepatectomy for HCC are "unclear". Actually it was "very clear" if you check from the references.
3. RNA sequence and CyTOF were obstained from non-cirrhosis HCC, how about the cirrhotic HCC?
4. Please add more descript comcerning about the differnt results from either the cirrhotic and non-cirrhotic liver , or HBV and HCV oatients after hepatectomy in the "Discussion"
5. Please add a "conclusion" in the last of the "Discussion"
6. Please try to translate the references in Chinese into English
Round 2
Reviewer 2 Report
none